# *Reticulitermes flavipes* (Blattodea: Rhinotermitidae) Response to Wood Mulch and Workers Mediated by Attraction to Carbon Dioxide

**DOI:** 10.3390/insects16020194

**Published:** 2025-02-11

**Authors:** Tae Young Henry Lee, P. Larry Phelan

**Affiliations:** 1Department of Entomology, College of Food, Agriculture and Environmental Sciences, The Ohio State University, Wooster, OH 44691, USA; lee.t1@ufl.edu; 2Department of Entomology and Nematology, University of Florida, 1881 Natural Area Drive, Gainesville, FL 32611, USA

**Keywords:** *Reticulitermes flavipes*, subterranean termite, food finding, carbon dioxide, attraction

## Abstract

The global economic impact of subterranean termites is estimated to top USD 30 billion annually in damage to wood structures and costs of control measures. For insects living in the soil, resource finding brings special challenges not faced by above-ground species. These challenges include the limited cues available to provide information about the resource, physical obstructions to their detection in the soil, and the high energy costs of tunneling. The termite *Reticulitermes flavipes* has evolved a locomotory program of steering that increases the search area and thus the probability of an encounter with potential food substrates. Nevertheless, the integration of a directional response to resource-associated cues is predicted to further increase the energetic efficiency of this search strategy. In this study, we demonstrate that *R. flavipes* show attraction to moist wood mulch or mulch combined with termite workers. Solvent extracts of these materials elicited at best a partial response, while CO_2_, when presented at the concentration emitted by mulch, produced a termite response comparable to the mulch itself. Understanding the full suite of the mechanisms used by subterranean termites in finding resources for feeding and colonization is important for the development of new strategies for monitoring and control.

## 1. Introduction

The subterranean termite *Reticulitermes flavipes* (Kollar) employs an intermediate nesting type, which entails the occupation of multiple, spatially dispersed resources connected by a network of tunnels. Such a nesting habit requires both the continuous discovery of new resources and the redistribution of nestmates. Resource discovery engenders different challenges for subterranean insects compared to above-ground species. First, tunneling through soil is considered energetically more expensive than other forms of locomotion [1] and varies with soil texture, moisture level, and other abiotic factors. Thus, evolutionary theory would predict selection for mechanisms that maximize the efficiency of finding resources. Second, soil-dwelling organisms are limited in cues useful for food finding, e.g., the common use of visual cues and wind are excluded, the latter of which transports odor molecules and indicates the direction of their source, even in the absence of a concentration gradient. In addition, for subterranean termites, a tunnel construction that maximizes search efficiency must be balanced with the need to optimize food transport back to the nest [2]. Consistent with the predicted selection for higher food finding efficiency in tunneling termites, many studies have documented the use of a self-steered idiothetic algorithm in the absence of allothetic resource cues that governs search tunnel geometry in subterranean termites [2,3,4,5]. While the initial direction of tunnel construction is random, termites use path integration, possibly mediated by a magnetic sense [6], to create tunnels that are relatively straight, following what is termed a Global Away Vector [7]. The collective result of multiple explorer termites is a pattern of tunnels radiating from a central origin without backward looping. Straight primary tunnels combined with an optimal ratio of branched secondary tunnels increases the efficiency of finding clumped food resources as utilized by termites in nature [5].

This self-steered search strategy of termites does not, however, preclude the use of resource-associated cues to aid in food finding. For most motile organisms, search for resources is characterized by an initial phase of extensive exploration to maximize the area covered, where movement is governed by idiothetic mechanisms and/or geographically relevant cues [8,9]. This phase is followed by a second phase of intensive localized search, triggered by cues associated with the resource. Organisms optimize search efficiency by balancing these two strategies based on information availability and resource uncertainty [9]. Switching to an area-restricted search where orientation to food cues increases search efficiency by expanding the “active space” of the resource that can be intercepted [10].

Research on directed orientation by subterranean termites to food resources has drawn mixed conclusions. Several reports suggest a role for chemical cues in termite food discovery, including decaying wood and extracts, feeding stimulants, and CO_2_ [11,12,13,14,15]. However, other studies find no evidence for orientation to chemical resource cues [16,17,18,19]. Also unresolved, even where a response to cues is suggested, are the orientation mechanisms employed to locate the source and the medium transporting the cues for food discovery. Kennedy [20] distinguishes attraction as directed movement oriented toward the source via taxes, from arrestment, defined as undirected movement involving kineses. Thus, termites might be found on a resource because they were attracted from a distance by cues or because their movement became arrested by those cues only after they arrived at random or by other locomotory mechanisms. Since most bioassays testing termite response to cues allow for free movement between treatments, conclusions cannot be drawn about the proximal mechanisms mediating preference [14,15,21,22]. There are two main channels for the transport of chemical cues for soil organisms: chemicals can travel through water or gas in the pore space between soil particles. Cornelius and Lax [14] concluded that the water-soluble components of a preferred substrate leaching into surrounding soil directed foraging behavior by *Coptotermes formosanus*, whereas Reinhard et al. [23] recorded a directional response to wood by *R. santonensis* both in soil and in the open, concluding that the response was mediated by wood volatiles.

Here, we tested the response of foraging *R. flavipes* to the polar and nonpolar extracts of rotting wood mulch and nestmates, and to CO_2_ using a bioassay design that distinguished attraction from arrestment in response to airborne cues. We hypothesized that *R. flavipes* would show attraction directed by volatile agents emanating from food sources and nestmates.

## 2. Materials and Methods

### 2.1. Termite Colonies

Termites were taken from the lab-reared colonies of *R. flavipes* established between 2004 and 2010 in Ohio. Colonies were maintained in 24 hr dark at 27 °C and 60% RH, and were regularly supplied with pine blocks and Vigoro^®^ premium brown wood mulch. Termites were collected from the colony by placing a PVC pipe (5.5 cm diam × 9 cm long) containing moistened corrugated cardboard rolls in the colony. Termites were then gently removed from the rolls and immediately placed into bioassay arenas for testing.

### 2.2. Y-Tube Olfactometers

#### 2.2.1. Exit Olfactometer

A Y-tube olfactometer design allowing for unrestricted movement among chambers was used first to observe termite response to food and nestmates (Figure 1). The olfactometer consisted of three chambers: a termite introduction chamber and two treatment chambers (Figure 1a). The introduction chamber was a cylindrical plastic container (3.6 cm high × 5.2 cm diam) (Figure 1b). Each treatment chamber was composed of two 50 mL centrifuge tubes with cone bottoms removed and joined, and metal screen installed between (10 cm × 3 cm; L × diam; Figure 2a). All chambers had one 0.5 cm diameter hole at their base through which Tygon^®^ tubing (4.5 cm long × 0.5 cm OD) (Poznań, Poland) was passed, allowing for the egress of termites. A 0.5 cm OD glass Y tube with 5.5 cm arms connected the chambers via the Tygon^®^ tubing. The center of the Y tube had a vertical opening (0.9 cm diam) with a screw-on cap lined with PTFE tape (Figure 1c). The arms of the Y tube had a broader opening distally to accept the Tygon^®^ tubing and the same inner diameter (0.5 cm) as the tubing.

One of the treatment chambers contained workers with a water-saturated glass microfiber filter (21 mm diam grade GF/A, Cytiva Whatman^TM^, Fisher Scientific (Pittsburgh, PA, USA)), moistened mulch (0.6 g), or both workers and mulch, while the control chamber contained a water-saturated glass microfiber filter only. In this and subsequent experiments, the positions of the test and control chambers were randomized between replicates.

#### 2.2.2. No-Exit Olfactometer

The second Y-tube olfactometer was designed to determine whether the termite response was a product of attraction or arrestment. This design was the same as the “exit olfactometer”, except that the entrance was elevated by 1.5 cm and fitted with a 1 mL pipette tip (2.5 mm diam; Figure 2b). This modification was the result of testing different configurations, in which it was demonstrated that termites were not deterred from entering the treatment chamber but were prohibited from leaving. After the initial experiment comparing termite response in this design with the design allowing for free movement out of the treatment chambers, the no-exit modification was used for all subsequent experiments.

One of the treatment chambers (test chamber) contained workers with a water-saturated glass microfiber filter (21 mm diam), moistened mulch, or a combination of mulch and workers (without glass microfiber filter), while the control chamber contained a water-saturated glass microfiber filter only. Three doses of the treatments were tested: 0.6 g, 1.2 g, and 1.8 g of mulch; 30, 60, and 90 *R. flavipes* workers.

### 2.3. Olfactometer Assays

Behavioral bioassays were conducted at 27 °C, >65% RH, and continuous darkness. Thirty workers (third-instar or higher) were placed into the introduction chamber, with agar droplets on the lid to increase humidity (Figure 1b). A 5 cm binder clip was placed on the Tygon^®^ tubing connected to the introduction chamber to prevent termites from accessing the Y tube for 24 h. Prior to release, the screw cap of the Y tube was temporarily removed and air was pulled into the center using a pipette bulb to ensure an odor gradient had been established from the treatment chambers. The binder clip was then removed and the termites were granted access to the Y tube for 24 h, when the number of termites in each chamber was recorded. The Y tubes and plastic containers were rinsed with a 10% acetone solution and cleaned using Alconox^®^ (New York, NY, USA) between experiments. The Y tubes were also oven-heated at 120 °C prior to use in subsequent experiments, and Tygon^®^ tubing was replaced after each use.

#### 2.3.1. Response to Mulch Extracts

One of two treatment chambers (test chamber) contained a glass microfiber filter (21 mm diam) treated with 2 mL of hexane, acetone, or the distilled water extract of mulch (equivalent to 1 g of mulch), while the control chamber contained a glass microfiber filter treated with 2 mL of solvent. The extracts and solvent blanks were applied in 0.5 mL aliquots, allowing for time in between for the solvent to evaporate. The treated filters were separated from the test subjects by a metal screen mesh. The response to extracts was compared to a positive control, in which the test chamber contained 1.2 g of moistened mulch and the control chamber contained a water-saturated glass microfiber filter.

#### 2.3.2. Response to CO_2_ Removal

The treatment chamber was modified to test CO_2_ removal on termite response to the source materials. A 2.5 cm gap was created at the center of treatment chambers, either side of which was a metal screen mesh (Figure 2c). The gap of both the test and control chambers was filled with soda lime (CO_2_ removal), spent Drierite™ of similar granule size (physical blockage), or nothing (positive control). As previously stated, the distal end of the test chamber contained the following treatment: workers (*n* = 60) with moistened glass fiber filter paper, moistened mulch (1.2 g), or both.

#### 2.3.3. Response to CO_2_ Concentrations

The test and control chambers were provisioned with 10 g of moistened sand (25% by weight) and the distal lid held a rubber septum through which the needle of a 500 mL syringe could be inserted (Figure 2d). Response to CO_2_ concentrations ranging from 1300 to 430,000 ppm were tested by injecting a calculated volume of CO_2_ into the syringe, then filling the remaining volume with ambient air. Prior to each test, a 10 µL gas sample was collected from the syringe to measure the actual CO_2_ concentration and air from the treatment chambers was pulled into the center of the Y tube using a pipette bulb. As a positive control, termite response to 1.2 g of mulch + 60 workers was measured to compare with response to CO_2_ alone. Prior to each bioassay, 10 µL gas sample was collected from the treatment chambers to determine the CO_2_ concentration generated by the termites and mulch.

The concentration of CO_2_ was determined using an Agilent Technologies 7890A gas chromatograph interfaced to a 5975C inert mass selective detector (Santa Clara, CA, USA). A 10 µL gas sample was manually injected into the gas chromatograph inlet in splitless mode at 25 °C. The GC column was J&W DB-5, 30 m × 320 µm × 1 µm film thickness. Helium flow was set at a constant 1.3 mL/min. The oven was held at 25 °C, for a total run time of 5 min. The MS was operated in the SIM mode using *m*/*z* 44 and peak areas were recorded. Concentrations were calculated by comparison of peak areas to a calibration curve generated each day using five gas samples of known CO_2_ concentrations.

### 2.4. Statistical Analysis

Statistical analysis was conducted in R 4.1.1 (R-project.org, 2021). Termite response to treatments in exit and no-exit bioassay was measured as the Response Index (RI), according to the formula

RI = (T − C)/30,

where T and C are the numbers of termites at 24 h in the test and control chambers, respectively, and 30 indicates the total number of termites in a bioassay arena. After testing for normality, a single-sample *t*-test was used to test the hypothesis that the RI was not different from 0. An ANOVA was conducted to compare termite responses among treatments in the extract experiments. If the ANOVA indicated a significant effect, least-significant difference was conducted to identify the differences.

For the CO_2_ removal experiment, the normality of RI could not be achieved with transformation, so the Kruskal–Wallis test was employed to compare the numbers of termites found in the test and control chambers for each treatment and to test for the differences among CO_2_ removal treatments in the termite numbers found in the test chamber.

Regression analyses of response to graded treatments were conducted using a fitted line plot, starting with a quadratic model. To calculate variance in the predicted CO_2_ concentration optimum that elicited the highest response, the data were bootstrapped with 10,000 replications.

## 3. Results

### 3.1. Response to Mulch and Nestmates

When responding *R. flavipes* workers were allowed unrestricted movement in and out of treatment chambers, significantly higher numbers were elicited by mulch alone (t = 3.65, df = 9, *p* = 0.005) or mulch + workers (t = 3.24, *p* = 0.01, df = 9) compared to the control, but not by workers alone (t = 1.68, df = 9, *p* = 0.13) (Figure 3). On average, about twice as many termites were found in the test chambers containing mulch compared to the control chambers, and over thrice as many termites gathered in the test chambers containing mulch + workers compared to the control. This pattern of activity indicated that the active cues were unique to the mulch or that the workers alone simply produced subthreshold levels of cues. RI was nearly identical when egress from the treatment chamber was precluded (Figure 3), demonstrating that termite response was mediated by attraction to air-borne cues. This design was used for all subsequent bioassays.

### 3.2. Attraction and Treatment Dose

Termites showed a significant response to mulch at all dose levels (RI @ 1×: t = 4.76, df = 9, *p* = 0.001; 2×: t = 6.30, df = 9, *p* < 0.001; 3×: t = 3.76, df = 9, *p* = 0.004); however, neither linear nor quadratic fitted line analysis indicated a significant dose effect (Figure 4a). In contrast, response to workers showed a significant dose-dependent pattern (r^2^ = 0.94, *p* = 0.01): at 1×, RI was not significantly >0 (t = 1.73, df = 9, *p* = 0.12), but increased at higher doses (2×: t = 14.24, df = 9, *p* < 0.001; 3×: t = 9.13, df = 9, *p* < 0.001) (Figure 4b). Like mulch alone, mulch + workers elicited significant attraction at all doses (1×: t = 3.24, df = 9, *p* = 0.01; 2×: t = 4.21, df = 9, *p* = 0.002; 3×: t = 10.67, df = 9, *p* < 0.001) and only a nonsignificant trend for the dose effect (Figure 4c). The results of this experiment indicate that the nonsignificant response to workers alone in the preceding bioassays was due to insufficient levels, while the similar non-dose-dependent response to mulch with or without workers suggests the two may share the same active volatile cue.

### 3.3. Response to Mulch Extracts

Termites did not show significant response to either acetone or dH_2_O extract of mulch (acetone: t = 0.83, df = 23, *p* = 0.41; dH_2_O: t = 0.11, df = 15, *p* = 0.91) (Figure 5a,b). Significant response to hexane extracts of mulch was recorded in both experiments [t = 3.14, df = 7, *p* = 0.02 (Figure 5a) and t = 3.01, df = 22, *p* = 0.006 (Figure 5b)]. However, hexane extract elicited termite response significantly lower than to intact mulch in the dH_2_O experiment (Figure 5b), with a similar 2× lower response in the acetone experiment that was not statistically different from intact mulch (Figure 5a).

### 3.4. Response to CO_2_

Termites never showed a significant preference for the test chamber when CO_2_ was removed from the emissions of the test materials using soda lime (Figure 6). Soda lime significantly reduced the number of termites locating in the test chamber relative to the blank for the treatments containing mulch. As in previous experiments, workers alone did not elicit a significant response even in the absence of soda lime. In the presence of spent Drierite™, which was intended to control for the physical blockage of gas movement by soda lime particles, a significantly higher number of termites in the test chamber was observed compared to the control chamber for all three test materials (Figure 6). Compared to the blank, the number of termites in the test chamber was not significantly different with Drierite™ when testing workers alone or mulch + workers, but was lower when mulch was tested. Thus, it appears that the removal of CO_2_ by soda lime eliminated orientation to attractive materials. Although there was evidence that physical interference by Drierite™ may also have reduced response, this effect was insufficient to explain the loss of response caused by soda lime.

To determine whether CO_2_ alone was sufficient to account for termite attraction to the treatments, termite response to mulch + workers was compared to a range of CO_2_ concentrations. Termite response to CO_2_ concentrations yielded a quadratic model (RI = −52.6x^2^ + 435.5x − 859.7, where x= log(CO_2_ ppm) with a peak response predicted at 13,849 ± 2528 ppm (Figure 7). The mulch + workers treatment was determined to produce CO_2_ emissions of 10,125 (±3162) ppm on average, eliciting a termite RI of 0.29 (±0.21). This result is not different from the RI = 0.41 predicted by the model for CO_2_ alone at the same concentration.

## 4. Discussion

Here, we found that foraging *R. flavipes* oriented to rotting wood mulch, either alone or in combination with termite workers, and that this behavioral response constituted attraction as a directed orientation to odor from a distance and not due to differential arrestment after contact. Overall, termites showed significantly greater response to mulch than to termites at the dosages tested, and combining the two did not significantly increase the response. When different amounts of the three treatments were presented, termites showed higher response with an increase in the number of workers as the source, whereas they showed no change in response to different amounts of wood mulch and a nonsignificant increase in response to mulch + workers.

We found some attraction to the hexane extracts of mulch, although this response was significantly lower than that to intact mulch. By contrast, there was no evidence that attraction was mediated by volatile constituents extractable by polar solvents. In subsequent experiments, the removal of CO_2_ from the headspace of mulch, workers, or mulch + workers resulted in a complete loss of termite response. Furthermore, termites showed a quadratic response to CO_2_ concentrations with a maximum response at ca. 14,000 ppm, while the mulch + workers treatment released CO_2_ levels of ca. 10,000 ppm on average. Combined, these results suggest that CO_2_ is both necessary and sufficient for a full response to mulch + workers in our olfactometer. These results are consistent with the findings of Bernklau et al. [12], who demonstrated a maximal response to 5000–10,000 ppm CO_2_ by *R. flavipes*, *R. tibialis*, and *R. virginicus* using a bioassay design similar to ours. We extended these past findings by demonstrating that CO_2_ was both necessary and sufficient to explain the orientation to our rotting wood mulch. The weaker response to hexane-extractable cues from mulch appeared to be redundant rather than synergistic with CO_2_. These results do not rule out termite response to other chemicals not emitted by the rotting mulch or the termites we tested. Also, since the bioassay was designed to test response to volatile cues, it does not address the potential role of chemicals moving through the soil solution.

### 4.1. Foraging Behavior

The initial localization of food resources by most organisms using an active search strategy is composed of three stages: (a) non-resource-directed search behavior, (b) resource-cue-directed orientation, and (c) resource assessment [24]. The first stage is characterized either by random locomotion or a self-steered program of movement that increases the efficiency of encountering the resource prior to detecting any resource cues. Optimal foraging theory would predict that when food is clumped in distribution, as is the case for subterranean termites, locomotory patterns should minimize the time and energy needed to move from one patch to the next, using an extensive search mode characterized by relatively straight movement [25]. For many above-ground insects, a locomotory search program commonly expresses as a zigzagging or looping motion of increasing size [26], resulting in an ever-expanding sampled space. Multiple studies have demonstrated an extensive search phase in subterranean termites mediated by a self-steered algorithm that in the absence of obstructions creates a pattern of relatively straight tunnels [27,28], with efficient displacement away from the origin point. Fractal analysis of the simulated tunnel architecture, based on the parameters identified by Su et al. [7], affirmed that the causative search geometry was optimal for encountering food in a patchy distribution [29].

Some question the role of chemical cues for food discovery by subterranean termites, suggesting the extensive search pattern of termites is sufficient and citing the slow movement of chemicals below ground [28]. However, as argued by Bartumeus et al. [8], search efficiency is measured by the discovery of targets not by the space sampled. Thus, an extensive search pattern in insects is usually accompanied by an intensive search, triggered by resource-associated cues, where movement is more tortuous, and searching more localized [25]. Search efforts mediated by cues are made more efficient than random encounters by expanding the detectable active space of the resource and thus increasing the probability of its discovery. A difference in active space for subterranean termites was demonstrated by Puche and Su [18], who documented tunneling as near as 2.5 mm from a disk of sound wood without finding it, whereas decaying wood elicited orientation from 12 to 18 cm away [13]. Attraction to saprophytic fungi-infected wood may be adaptive, as *R. flavipes* can show higher survivorship and behavioral preference for this substrate compared to sound wood, depending on the level of decay and tree species [11,30].

### 4.2. Response to Volatiles in Soil

Early researchers of host finding by root herbivores also promoted random encounter as the predominant mechanism [31]. Subsequently, orientation to a broad range of chemical cues by below-ground organisms has been demonstrated, with CO_2_ eliciting a strong response in the majority of soil-dwelling insects, either alone or in combination with other volatiles [32,33,34]. The soil environment is well suited to CO_2_ as a reliable cue of source direction compared to the above-ground setting [35]. The dilution and dispersal of CO_2_ in the open atmosphere is dominated by turbulence and fast-moving air, which destroy a concentration gradient and directional information over a short distance. By contrast, in the relatively static conditions of soil, CO_2_ transport is regulated predominantly by molecular diffusion [36], coupled with reduced diffusion rate coefficients, 0–45% of that in open air, depending on the level of air-filled porosity [37]. This combination creates conditions that favor higher concentrations and steeper gradients of gases and volatile chemicals from their source [38]. It has also been argued that CO_2_ is too ubiquitous in the soil to be a useful signal of resources [35]. However, if orientation is mediated by a concentration gradient and not just the presence of CO_2_, then directionality can be encoded even at high background levels of the stimulus. In some cases, response to CO_2_ may be supplemented by other volatiles to distinguish preferred targets. *Diabrotica virgifera virgifera*, a specialist herbivore of maize roots, responds to numerous root volatiles, but most strongly to CO_2_. Larvae of which the CO_2_ receptors were silenced were able to locate maize roots when released within 9 cm, but not when tested from longer distances. This result suggested that CO_2_ may act to draw larvae from longer distance, but at closer proximity, other compounds allow larvae to distinguish the preferred host from nontarget sources of CO_2_ [39]. Compared to nonsocial below-ground herbivores, subterranean termites have the additional challenge that nests generate high levels of CO_2_, with the potential to trap termites from ever leaving the nest. However, this conundrum could be explained by the self-steered program underlying Global Away Vectoring, which governs the initial movement of exploratory termites and blocks looping back to their origin.

Organisms can use two sets of orientation mechanisms in their search behavior, kineses and taxes, depending on cue structure and organismal sensory capacity [40]. Taxes represent directional responses to cue intensity, where the organism makes simultaneous comparisons between paired receptors or sequential comparisons in time as it moves through its environment. Taxes are the more efficient mechanisms when cue structure provides clear information about source location. Kineses are less dependent on cue structure as an indicator of direction, and bring the organism into proximity with the source through increased turning and/or a reduced rate of linear locomotion with higher concentrations of the stimulus. Although the current study was not designed to quantify the parameters of locomotion, it is clear that the termite response was directional and not a result of differential arrestment after the source was encountered. Moreover, if the soil environment creates a cue structure that provides reliable information about the source location as suggested above, the more efficient, directional taxes would be favored.

## 5. Conclusions

In conclusion, the results of this study indicate a role for CO_2_ in the search for resources and nestmates by *R. flavipes*, using locomotory mechanisms of attraction. We suggest that arguing for a self-steered search pattern versus orientation to resource cues represents a false dichotomy and that both play important roles in increasing the efficiency of search and reducing the energy costs of tunneling underground. While the idiothetic mechanisms drive the initial stage of extensive search, maximizing the space sampled, intensive search by *R. flavipes* appears to be mediated primarily by CO_2_, which was both necessary and sufficient to explain orientation to the stimuli tested here. The widespread use of CO_2_ by soil-dwelling organisms may be driven by the creation of stable gas gradients by the soil environment which provide directional information for resources. We found evidence for a significant but weaker response to hexane-extractable volatiles, suggesting a possible secondary role for other chemical cues. Reinhart et al. [23] found no difference in response to volatile cues by *R. santonensis* when comparing walking in open air to tunneling in sand. Since our bioassay measured orientation to volatiles in open air, we cannot, without further experimentation, make the same claim for our findings, nor do we rule out additional response to water-borne cues in the soil. Also not clear from this or previous studies is how much CO_2_ expands the active space of the food resource in a natural context, within which directed orientation is triggered. This is hard to assess since the size and structure of the CO_2_ gradient in soil depend on numerous edaphic factors, such as texture and moisture level. Additional study is important to our fundamental understanding of subterranean termite search behavior, as well as its application for improved control. For example, how much does intensive search mediated by CO_2_ or other cues enhance the efficiency of resource location above the extensive search algorithm of termites? And do additional cues mediate resource assessment and subsequent colonization?

## Figures and Tables

**Figure 1 insects-16-00194-f001:**
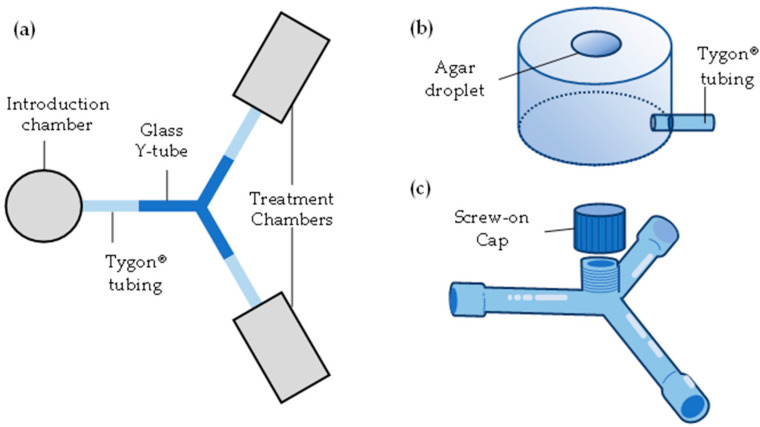
Y-tube olfactometer design (**a**), with details of termite introduction chamber (**b**) and glass Y tube (**c**).

**Figure 2 insects-16-00194-f002:**
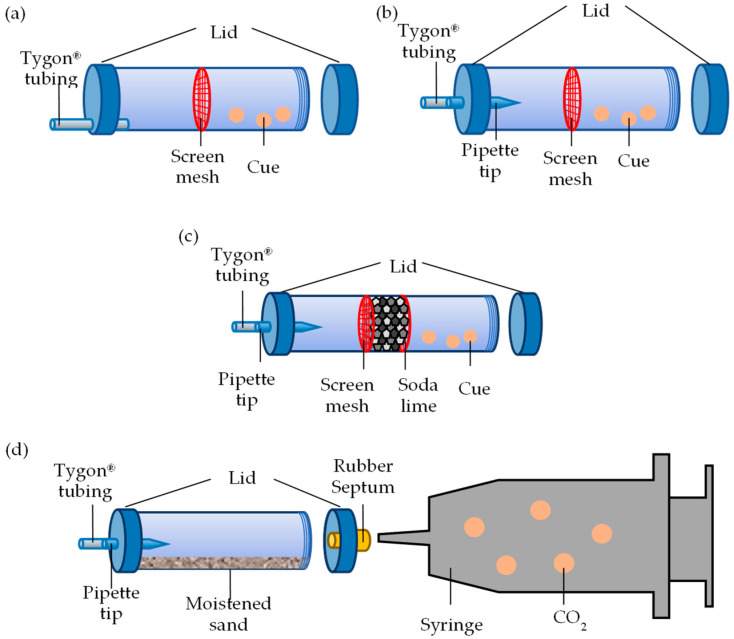
Treatment chamber designs: (**a**) exit olfactometer, allowing for unrestricted movement; (**b**) no exit olfactometer, preventing exit upon entry; (**c**) no-exit olfactometer with a gap to contain soda lime, Drierite™, or nothing; and (**d**) no-exit olfactometer connected to a 500 mL syringe containing CO_2_.

**Figure 3 insects-16-00194-f003:**
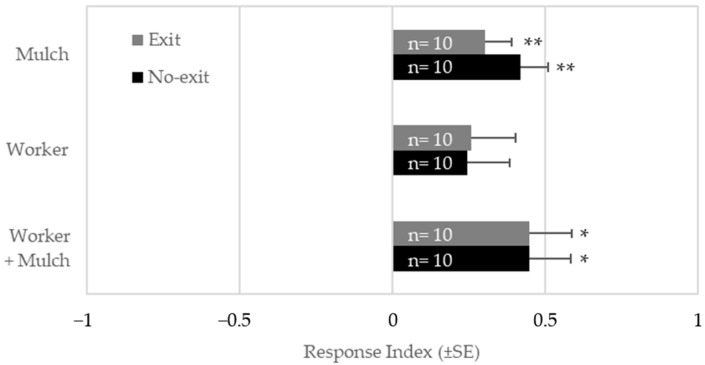
Response Index (RI) of termites presented with either a worker, mulch, or both by treatment and olfactometer design. The exit olfactometer allowed for unrestricted movement, whereas the no-exit olfactometer prevented exit from treatment chambers after entry. Asterisks indicate that the RI was different from 0 by a one-sample *t*-test (* 0.01 ≤ *p* < 0.05, ** 0.001 ≤ *p* < 0.01). Replication numbers (of 30 workers each) are shown within the bars.

**Figure 4 insects-16-00194-f004:**
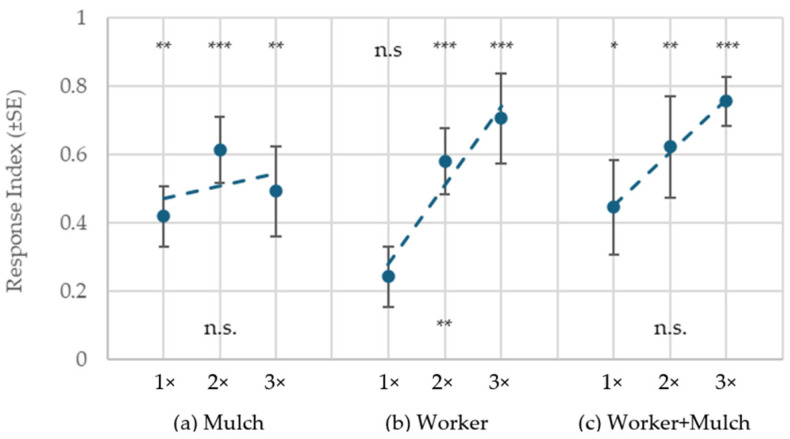
Response Index (RI) of termites presented one of three treatment doses, where 1× = 0.6 g mulch, 30 workers, or both. Asterisks on top indicate that the RI is significantly different from 0 by a one-sample *t*-test (* 0.01 ≤ *p* < 0.05, ** 0.001 ≤ *p* < 0.01; *** *p* < 0.001). Asterisks on bottom indicate significant linear regression of the dose effect.

**Figure 5 insects-16-00194-f005:**
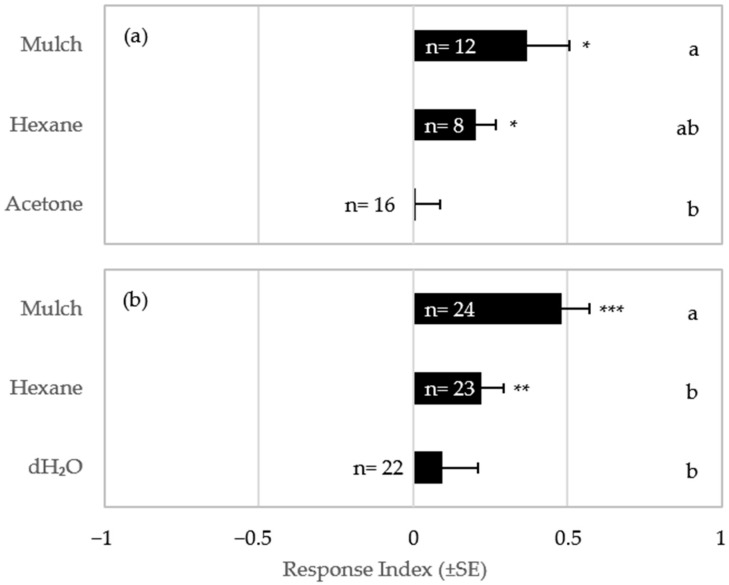
Response Index (RI) of termites presented wood mulch or the extracts of wood mulch: (**a**) hexane or acetone extracts; and (**b**) hexane or distilled water extracts. Asterisks next to each error bar indicate that the RI is different from 0 by a one-sample *t*-test (* 0.01 ≤ *p* < 0.05, ** 0.001 ≤ *p* < 0.01; *** *p* < 0.001). Different letters on the right indicate significant differences among treatments by protected LSD. Replication numbers (of 30 workers each) are shown within the bars.

**Figure 6 insects-16-00194-f006:**
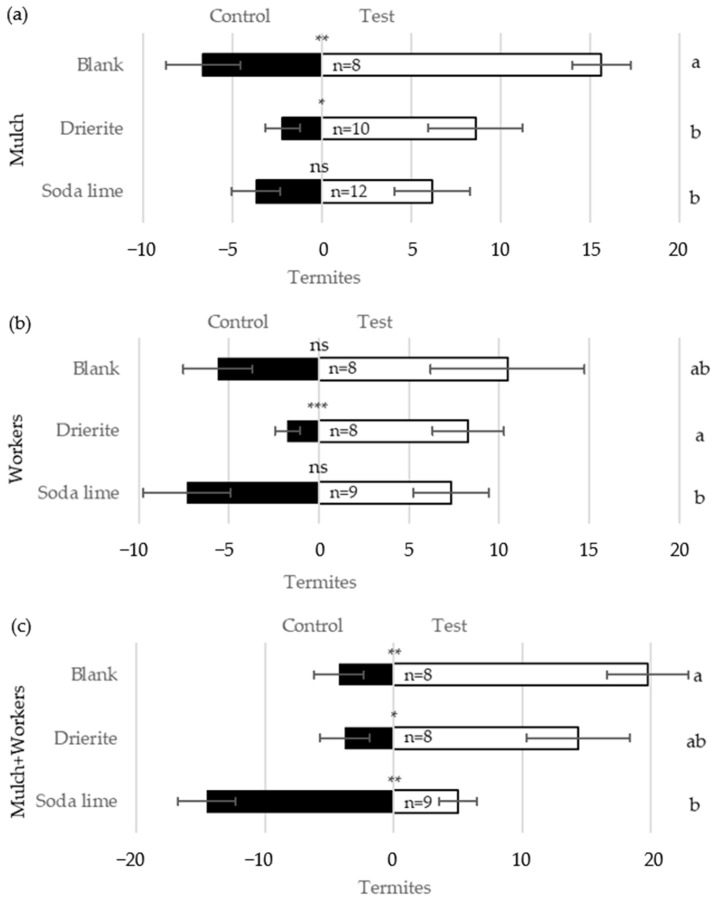
Mean number (±SE) of termites located in the Y-tube test chamber containing moist wood mulch (**a**), termite workers (**b**), or mulch + workers (**c**) compared to the control chamber. Middle section of the treatment chambers contained soda lime to remove CO_2_, spent Drierite™ as the control for the physical blockage of air movement, or nothing (Blank) as the positive control. Asterisks above bars indicate differences in termite numbers between chambers (* 0.01 ≤ *p* < 0.05, ** 0.001 ≤ *p* < 0.01, *** *p* < 0.001, or ns = not significant) measured by the Kruskal–Wallis test (note: nonparametrics were used since RIs could not be normalized for this experiment). Letters to the right compare termite numbers in the test chamber among CO_2_ removal treatments via the Kruskal–Wallis test. Replication numbers (of 30 workers each) are shown within the bars.

**Figure 7 insects-16-00194-f007:**
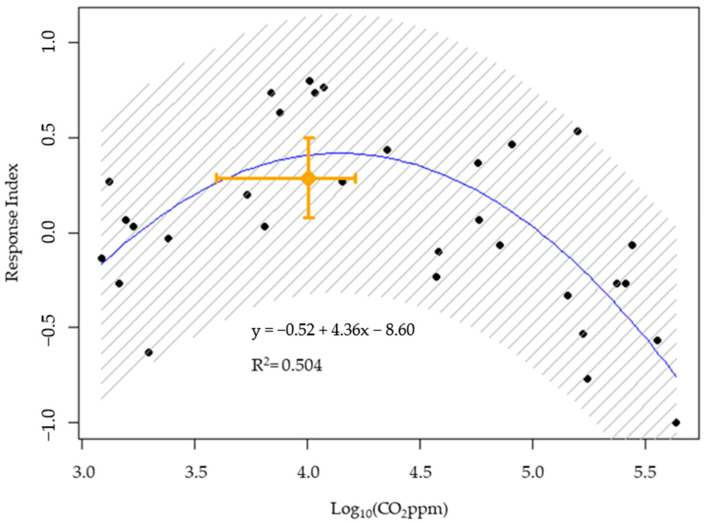
Response Index of termites presented CO_2_ concentrations from 1300 ppm to 430,000 ppm. The best-fit quadratic model is represented by the blue line with the greyed area encompassing the 95% CI. The orange point and bars represent the mean CO_2_ emissions (±95% CI) (*x* axis) and Response Index (±95% CI) (*y* axis) for 1.2 g wood mulch + 60 termite workers.

## Data Availability

The raw data supporting the conclusions of this study are available from the authors without undue reservation.

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
