# Peer review of "Reticulitermes flavipes (Blattodea: Rhinotermitidae) Response to Wood Mulch and Workers Mediated by Attraction to Carbon Dioxide"

_insects, 2025, doi:10.3390/insects16020194_

Round 1
Reviewer 1 Report
Comments and Suggestions for Authors
Comments: 
Underground pests face significant energy consumption challenges when searching for food resources. This study cleverly designed various methods to explore the unique mechanisms used by underground termites to explore large spaces, which is very interesting and of great significance for the management of underground termites. At the same time, rich experimental results were obtained. This study is very helpful for understanding the food search mechanism of underground pests. Termites can cause global economic losses, and understanding the full set of mechanisms used by underground termites in searching for foraging and colonization resources is crucial for developing new monitoring and control strategies. I suggest revising and supplementing the following problems before publishing this paper.
The main problems are as follows:
Why not use the same age for testing insects that are third-instar or higher, as inconsistent states can affect the experimental results and conclusions?
In 2.3. Olfactometer assays, did the test box and control box on both sides of the Y-shaped tube not switch positions after each experiment to eliminate the influence of position?
In addition, under what environmental conditions is the olfactory testing equipment placed, without sufficient description? Termites mainly move in dark environments, so is olfactory testing conducted in dark environments?
It is common knowledge that many termites prefer to eat wood mulch or decaying wood. This article only measured the behavioral response of termites to wood mulch. The focus should be on separating and identifying the volatile components released by wood mulch, and further screening to obtain which volatile components play a key role? I think this result is more meaningful and important.
The paper only includes the determination of termite extracts without specific determination of volatile compounds, which is difficult to match with the volatile compounds mentioned in the title.
Author Response
Comment 1: Why not use the same age for testing insects that are third-instar or higher, as inconsistent states can affect the experimental results and conclusions?
Response 1: The rationale for using the broader age range is that the worker and sexual (nymphal) lines deviate after the 2nd instar and foraging behavior begins with 3rd or 4th instar. The higher-instar stages are difficult to distinguish as they differ primarily by small differences in size. It is true that there is variability in task behavior among and even within different instars of termites and had we narrowed the worker instar range tested, we may have seen lower variability in response. We agree it would be interesting to if different instars respond differently. However, the time required to do so would have reduced the number of replicates or experiments conducted in the same time period and the conclusions would have applied more narrowly to specific instars rather than workers in general. It is most important to note that any increase in variability only reduces the sensitivity for detecting treatment differences and does not create statistical significance that does not exist.
Comment 2: In 2.3. Olfactometer assays, did the test box and control box on both sides of the Y-shaped tube not switch positions after each experiment to eliminate the influence of position?
In addition, under what environmental conditions is the olfactory testing equipment placed, without sufficient description? Termites mainly move in dark environments, so is olfactory testing conducted in dark environments?
Response 2: In this and the following comment, we appreciate that the reviewer identified oversights in our description of the bioassay. We have added text to the manuscript to indicate that the position of the treatment and control chambers was randomized and provided more complete description of testing conditions.
Comment 3: It is common knowledge that many termites prefer to eat wood mulch or decaying wood. This article only measured the behavioral response of termites to wood mulch. The focus should be on separating and identifying the volatile components released by wood mulch, and further screening to obtain which volatile components play a key role? I think this result is more meaningful and important. The paper only includes the determination of termite extracts without specific determination of volatile compounds, which is difficult to match with the volatile compounds mentioned in the title.
Response 3: The objective of the study was to determine whether termite workers use olfaction in their search for food resources and if so, what are the volatile constituents mediating the response. The reviewer is correct that moistened wood mulch is well-established as a preferred food resource, which is why it was chosen as the starting treatment for our study. However, as discussed in the paper, there is considerable debate in the literature as to whether termites find resources solely by an ideothetically guided pattern of tunneling or if orientation is also mediated by odor cues associated with the resource. Our initial experiments supported the latter, justifying a search for active volatile constituents. We started as the reviewer suggests by separating volatile constituents of mulch, into those extractable by a nonpolar solvent (hexane) or one of two polar solvents (acetone or water). We found that neither of the polar solvent extracts elicited any activity and that the hexane extract elicited only partial response. These results in combination with our findings that CO2 (when tested at the levels released by moistened mulch) was both necessary and sufficient to match attraction to mulch justifies the focus on CO2. This contradicts the reviewer’s contention that identifying other volatile constituents would be more meaningful since at best they only play a secondary role relative to CO2. We actually did conduct GC/MS analysis of the solvent extracts and identified some compounds, but given our bioassay results, concluded that behavioral testing of these compounds was not justified.
Although we conclude that our results contradict the reviewer’s argument that extractable compounds are more significant than CO2, we do understand the reviewer’s assertion that the wording of the title could be misleading. To correct this, we have changed the title and have also corrected phrasing in the abstract and elsewhere that might suggest we tested extracts of termites.
Reviewer 2 Report
Comments and Suggestions for Authors
insects-3428905
Reticulitermes flavipes…
Lee & Phelan
General
An exciting read, since the paper is well written and equally interesting.
My main comment concerns the conclusion that CO2 alone mediates attraction to wood mulch resources. The authors themselves mention that the bioassay is designed to test only volatiles and does not necessarily reflect a natural environment.
The discussion quotes ref #35 that CO2 is too ubiquitous to be useful resource signal by itself. Live plant roots, which are an inadequate substrate for termites, are mentioned as a source of CO2 in the discusion. I assume that even a dead rodent in a soil burrow would be strong CO2 source.
I'm not even entirely sure that a role of other volatiles can be excluded altogether. The intro claims (line 54) that an airflow underground can be excluded altogether. I disagree, odorants do travel through soil crevices (plenty of references in connection with attraction of larvae and nematodes in soil) and certainly also through termite tunnels.
This brings me to questions concerning the methods and/or the interpretation of the results obtained with the methods used.
Termite attraction to hexane extracts was lower than attraction to mulch (which is expected), but was anyhow different from blank? Do hexane solutions (non-polar solvent) contain CO2? And if so, possibly in very small amounts, for how long? It would be necessary to show that hexane solutions do release significant amounts of CO2 during the test - if the conclusion is maintained that CO2 is the only volatile cue attracting termites? Admittedly, it wouldn't be trivial to determine CO2 concentrations in hexane, but it would certainly be important to check that.
And, can we be sure that soda lime only removes CO2 and not any other possible/hypothetical cues?
This said, I have a hard time to understand the "Response Index", which is embarassing, but I just don't get it (line 202). If all termites (N=30) respond to the treatment, the RI becomes 100 [RI=100*(30-0)/30=100].
The scale for the RI in the figures is 1. And even if the multiplier (100) is omitted, I stil don't get it. Admittedly, probably my fault - but: why not just show the number of responding termites? That would be a simple and straightforward way of showing the results, and facilitate understanding. In this context - why, out of a sudden, does Figure 6 show the number of responding termites?
Author Response
Comment 1: My main comment concerns the conclusion that CO2 alone mediates attraction to wood mulch resources. The authors themselves mention that the bioassay is designed to test only volatiles and does not necessarily reflect a natural environment.
Response 1: We are not sure about this comment, but in combination with a later comment below, it would appear that the reviewer is misinterpreting ‘alone’ as ‘only,’ whereas it is used to indicate ‘by itself.’ This should be clear as we use the term many times when referencing other treatments, e.g., “workers alone” or “mulch alone.” It is impossible to conclude that a chemical is the only one that elicits a behavioral response without testing every chemical that exists. Instead, we conclude that CO2 (when presented by itself) is both necessary and sufficient to explain attraction to moist wood mulch and termites by two independent methods: 1) when CO2 is removed from mulch or mulch+workers using soda lime, attraction is lost (necessary) and 2) when CO2 is presented alone at a level matching that measured from mulch+termites, the level of attraction is the same as to mulch+termites (sufficient). By contrast, other volatiles, such as ones possibly mediating response to hexane extract, are neither necessary or sufficient to explain termite attraction to mulch and termites. We feel we clearly state the limits of our conclusions in the Discussion: “These results do not rule out termite response to other chemicals not emitted by the rotting mulch or the termites we tested. Also, since the bioassay was designed to test response to volatile cues, it does not address the potential role of chemicals moving through the soil solution.”
With regard to the comment that the bioassay may not reflect a natural environment, this can be said of all bioassays, which attempt to control the number of variables. By design, the bioassay only tests response to volatiles because we were interested in distinguishing olfactory-mediated attraction from orientation mediated by chemicals in the soil solution, a mechanism hypothesized by at least one previous study, as cited in the Introduction.
Comment 2: The discussion quotes ref #35 that CO2 is too ubiquitous to be useful resource signal by itself. Live plant roots, which are an inadequate substrate for termites, are mentioned as a source of CO2 in the discusion. I assume that even a dead rodent in a soil burrow would be strong CO2 source.
Response 2: We are not sure what the question is here. We suggest in the Discussion how a ubiquitous cue could still be useful for orientation. Also we provide multiple references of other organisms that use CO2 for food location.
Comment 3: I'm not even entirely sure that a role of other volatiles can be excluded altogether. The intro claims (line 54) that an airflow underground can be excluded altogether. I disagree, odorants do travel through soil crevices (plenty of references in connection with attraction of larvae and nematodes in soil) and certainly also through termite tunnels.
Response 3: This comment is incorrect as the reviewer is confusing molecular movement with air flow. We are in fact arguing for movement of volatiles in the soil, with the potential to mediate orientation of subterranean organisms. As explained in the Discussion with citations, it is well established that molecular movement in the soil is governed primarily by diffusion not by air flow. Nevertheless, to reduce confusion, we have changed ‘air flow’ to ‘wind’ to hopefully clarify the point.
Comment 4: Termite attraction to hexane extracts was lower than attraction to mulch (which is expected), but was anyhow different from blank? Do hexane solutions (non-polar solvent) contain CO2? And if so, possibly in very small amounts, for how long? It would be necessary to show that hexane solutions do release significant amounts of CO2 during the test - if the conclusion is maintained that CO2 is the only volatile cue attracting termites? Admittedly, it wouldn't be trivial to determine CO2 concentrations in hexane, but it would certainly be important to check that.
Response 4: Yes, as shown in Figure 5, in both the “Acetone” and “dH2O” experiments, the hexane extract elicited a Response Index value greater than 0 (as indicated by the * and **), but less than that for mulch (the difference was not significant in the Acetone experiment but was in the dH2O experiment, as indicated by the letters to the right).
With regard to CO2 in extracts, CO2 has similarly low solubility in hexane and acetone. More significantly, CO2 has substantially higher vapor pressure than any of the solvents used. Since we allow time to evaporate extract solvents prior to inserting into olfactometer, any CO2 in the extract would also be lost prior to testing. Thus, it is reasonable to assume that the response to hexane extract is not mediated by CO2.
As clarified in our response above, we are not stating that CO2 is the only volatile that can elicit attraction in termites.
Comment 5: And, can we be sure that soda lime only removes CO2 and not any other possible/hypothetical cues?
Response 5: This is a fair question to raise as it is an experimental assumption that the reduced response with soda lime was attributable to removal of CO2. This assumption is based on the fact that soda lime is widely used for purifying recycled breathing air for the expressed purpose of removing CO2. Other compounds removed include moisture, some nitrogen compounds, and some toxic acidic gases. However, with the exception of NO2, the extraction efficiency of most of these compounds by soda lime is much lower than for CO2, and all of these compounds, except H2O, are released at far lower levels than CO2 from decomposing wood. The role for CO2 suggested by this experiment was then confirmed by the positive response to CO2, when presented alone.
Comment 6: This said, I have a hard time to understand the "Response Index", which is embarassing, but I just don't get it (line 202). If all termites (N=30) respond to the treatment, the RI becomes 100 [RI=100*(30-0)/30=100].
The scale for the RI in the figures is 1. And even if the multiplier (100) is omitted, I stil don't get it. Admittedly, probably my fault - but: why not just show the number of responding termites? That would be a simple and straightforward way of showing the results, and facilitate understanding. In this context - why, out of a sudden, does Figure 6 show the number of responding termites?
Response 6: This comment identifies another error on our part, which we have corrected by removing the 100x multiplier from the equation. The Response Index is routinely used in binary choice bioassays, and is more preferred than comparing only termites in the test chamber because it accounts both for termites “responding” to the control chamber and for non-responding termites. As explained in the Methods, the CO2-exclusion experiment (Figure 6) had to be analyzed by the nonparametric Kruskal-Wallis test comparing termite #’s in test vs. control because we could not achieve a normal distribution of the RI for this experiment. Normality is an assumption of parametric tests such as t-test. We have made this decision more clear by adding an explanatory statement in the Figure 6 legend.